# Multicriteria Decision Making in Tourism Industry Based on Visualization of Aggregation Operators

Sergey Sakulin * and Alexander Alfimtsev

Institute of Informatics and Control Systems, Bauman Moscow State Technical University Scientific Research, 2-ya Baymanskaya st., 5, Moscow 105005, Russia; alfim@bmstu.ru
* Correspondence: sakulin@bmstu.ru

**Abstract:** The modern tourist industry is characterized by an abundance of applied multicriteria decision-making tasks. Several researchers have demonstrated that such tasks can be effectively resolved using aggregation operators based on fuzzy integrals and fuzzy measures. At the same time, the implementation of this mathematical tool is limited by weak intuitive understanding by the practicing specialists of the aggregation process as well as fuzzy measures in general. Some researchers have proposed different aggregation visualization methods, but these methods have several properties that block their wide implementation in decision-making practice. The purpose of this study is to develop a decision-making approach that will allow practitioners to have a clear intuitive vision of the aggregation process and fuzzy measures. This article proposes an approach to decision making in the tourist industry based on the synthesis of the aggregation operator that includes 3D visualization graphics in virtual reality. Firstly, some research devoted to decision-making methods in tourism was assessed along with "smart" tourism, aggregation operators and their visualization. Secondly, a 3D visualization in the form of a balance model was introduced. Thirdly, the method of aggregation-operator synthesis based on the 3D balance model and the 2-order Choquet integral was developed. Finally, an illustrational example of implementing such an approach for resolving the task of assessing and choosing a hotel was described.

**Keywords:** tourism industry; Choquet integral; fuzzy measure; visualization of aggregation operators; balance model; virtual reality

## 1. Introduction

The modern tourist industry is characterized by a multitude of complicated practical decision-making problems based on the evaluation of multiple criteria [1]. Such problems include group-tour planning, choice of service providers, hotel choice, etc. As decision makers, we have travel companies and agencies, hotel managers and tourists. The criteria include the specific requirements for the trip, provider or hotel. The "smart tourism" concept envisions the use of information and communication technologies for such tasks, actual data awareness and personalization [2].

In the current state of information overload in "smart tourism", an important place is devoted to expert recommendation systems by which users can make decisions based on their preferences. Such systems are built on multicriteria decision-making methods. The main purpose of these systems is to propose a suitable service while taking into account the personal preferences of an individual user. This allows the wellbeing of people to be improved by offering more interactive travel of a higher quality [2,3]. Despite the abundance of practical realizations in the sphere of decision making, tourist preferences need a finer degree of segmenting [4]. Such segmentation of preferences could be described by cultural, generational, geographical and other differences between the consumers of tourist services. Specifically, some tourists prefer personal communication with service providers and are not inclined to use the services rendered by intellectual systems like

the unmanned ground vehicle [4]. For such segmentation of tourists' preferences, "fine" decision-making methods can be applied.

Decision-making methods in the tourist industry rely on several theoretical approaches. One of these approaches is the use of aggregation operators [1]. In particular, the research in [5] describes the implementation of a multiattribute utility theory along with the Choquet integral for travel planning. Recently, the authors of [6] proposed new aggregation operators based on Pythagorean cubic fuzzy sets for decision making. The authors of paper [7] describe the use of extended families of OWA operators for efficient sentiment analysis of the text reviews of tourists.

The Choquet integral has a high degree of expressiveness as it allows the different interactions of the criteria to be represented. At the same time, there are practical difficulties in working with it [8].

This is why, in certain cases, in order to make a decision, a simplified approach based on a weighted average as an aggregation operator is implemented. Such a situation is explained by the cognitive difficulties of the decision maker during an aggregation-operator synthesis for multiple criteria. At the same time, the task of aggregation-operator synthesis with predefined parameters is solved with different visualization methods [9–11] but because of cognitive difficulties, these methods have not become widely available.

The present paper suggests an approach to multicriteria decision making that is based on the aggregation-operator synthesis including 3D visualizations of cognitive graphics. The process of aggregation-operator synthesis consists of two procedures. These procedures represent sets of actions carried out by the decision maker where at every step an action is required. An example of this approach is illustrated by the hotel choice decision-making task in a "smart city" from several alternatives. The advantages of the proposed approach are as follows. First, it brings practical improvements in the formalization of expert preferences through the implementation of a virtual object that reflects the aggregation operator. Secondly, this approach can be applied with standard virtual reality software and hardware for the synthesis and verification of aggregation operators for decision making.

The paper is further organized as follows: Section 2 deals with other research dedicated to decision-making methods and tourist recommendation systems together with aggregation operator visualization. These works were used to choose the corresponding components for building our approach. Section 3 describes the proposed visualization of aggregation operator setup. Section 4 is about the synthesis of the aggregation operator on the basis of the implied visualization. Section 5 shows a case of applying the suggested approach for making decisions in one tourism-industry sphere. Section 6 contains a discussion about the features of our approach from several points of view. Section 7 presents the main conclusions of the paper.

## 2. Related Works

Tourist recommendation systems are becoming widespread these days [12]. Some of such system's components are aggregation operators. The aggregation operator reflects the decision maker's opinion about the merged actions of several criteria. The simplest such operator is the weighted arithmetic mean [13]. Other operators like the OWA can be used [14]. The OWA operator and the weighted arithmetic mean are special cases of the Choquet integral.

There are many publications devoted to the implementation of the Choquet integral for tourist preference formalization, including hotel assessment [15,16], group preference identification [17] and restaurant ranking [18].

On the other hand, the amount of theoretical research on aggregation operators is growing exponentially [19], as well as the rapid development of practical methods for using aggregation operators, including the direct consideration of observable properties of human reasoning [20]. Many studies of aggregation operators are devoted to fuzzy measures and integrals, due to their flexibility and universality [21]. One of the most developed operators

of this class is the Choquet integral because it has a convenient practical application for "interaction representation" [22].

The implementation of the Choquet integral allows the employment of the expert's knowledge about criteria interconnections, including positive and negative correlations and substitutiveness (complementarity). The main obstacle towards wide practical usage of such aggregation operators is the difficulty of working with the decision maker on preference formalization. These complications are linked with poor intuitional vision of the aggregation process in general and the definition of fuzzy measures in particular.

In order to overcome such difficulties, some methods of aggregation operator visualization are being developed. One of these methods was based on the balance model [9]. It was used to create operators with this method, as an example of the functional dependence of a scale attached to a lever. As well as that, methods for visualizing Choquet integrals were introduced. Specifically, the graphical interpretation of the Choquet integral considers plotting on the coordinate plane a limitation line for the interaction indices and Shapley indices of the two criteria [10]. This idea was further developed in research [11] where the methods of fuzzy measure identification were developed and were based on a hierarchy diagram of diamond pairwise comparison using graphical interpretation. Research [23] suggests a second order Choquet integral visualization implementing the balance model. This visualization comes from a mutually exact comparison of a mathematical object (Choquet integral) and a physical object (a lever fixed in the center by a spring with a stiffness coefficient equaling one that can rotate around the horizontal axis). This physical object is well understood by humans who have an intuitive vision of it because of experience and physical intuition.

The aggregation visualization methods described above encounter certain difficulties. Namely, the fact that the graphical interpretation [10] is limited by having only two criteria and the hierarchical diagram in its current state does not allow one to "envision" the entire aggregation operator. The two-dimensional balance model [23] is not limited by two criteria but, in the case of multiple criteria (starting from about four to five—considering the added weights coordinated by interaction indices), the decision maker starts having trouble with a 2D visualization because the pictures of weights corresponding with the interaction criteria and indices can overlap and cover each other. Apart from that, none of the mentioned visualization methods allow us to model fuzzy measures higher than second order that reflect the dependencies of more than two criteria.

In recent years, there has been a rise in virtual reality applications that provide the opportunities to model different objects in 3D environments. Research [24] deals with the application spheres of 3D simplexes in decision making and education. Three-dimensional graphics and virtual reality possess broader capabilities for visualization in comparison with two-dimensional graphics, as the third dimension provides more information. And it is more natural for a human to perceive data and knowledge through 3D visuals. Virtual reality holds strong potential and has been applied in recent years with growing intensity, especially in higher education [25]. At the same time, as far as is known to the authors, aggregation operator visualization based on the balance model in a 3D virtual environment has never been realized before.

As a result, in order to test our ideas, we chose 3D graphics consisting of the 3D balance model together with the Choquet integral.

## 3. Visualization of Aggregation-Operator Synthesis with 3D Graphics Implementation

Let us analyze an existing Choquet integral visualization in the form of a 2D balance model [23]. This visualization presents an absolutely stiff lever fixed in its pivot point with a spring (Figure 1).

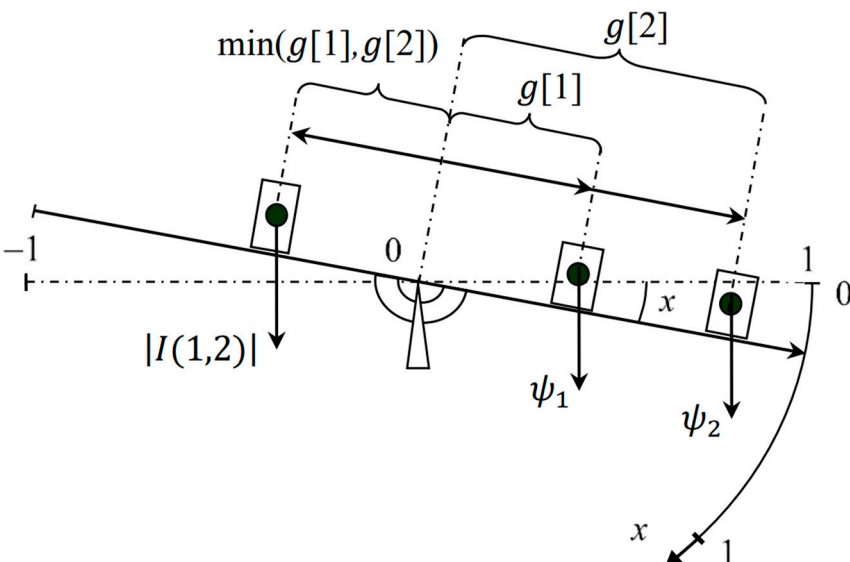

**Figure 1.** Two-dimensional balance model.

Weights are placed on the lever and correspond with the importance or "weights" of the criteria. In the non-negative part of the lever's scale criteria, weights $\psi_1$, $\psi_2$ are placed at $g[1]$ and $g[2]$ distances. In the negative part of the scale, we place weight $|I(1,2)|$ in coordination with criteria $g[1]$ and $g[2]$ interaction index at distance $\min(g[1],g[2])$ from the pivot point in the case that $I(1,2) < 0$. If the criteria interaction index is $I(1,2) \geq 0$, then weight $I(1,2)$ is added to criteria weight $\psi_1$ with the lowest value. Figure 1 shows the described balance between two criteria, the $I(1,2)$ interaction index of which is negative. In accordance with Newton's Second Law, the balance equation for this structure is $x = g[1]\psi_1 + g[2]\psi_2 + I(1,2)\min(g[1],g[2])$. The inclusion of more criteria to $H$ does not lead to changes in the balance structure:

$$x = \sum_{h=1}^{H} g[h]\psi_h + \sum_{\{i,j\}\subseteq J} I(i,j)\min(g[i],g[j]) \tag{1}$$

The above formula is equivalent to the 2-order Choquet integral where $J = \{1,\ldots, H\}$. The criteria value changes simultaneously change the lever position which in turn allows one to "see" the structure of the visualized aggregation operator. The transition to a three dimensional environment can be performed by upgrading the existing visualization (Figure 2).

The essence of this upgrade is that instead of a lever for criteria weight placement, we use a plane. Figure 2 shows a side view of this plane. An absolutely solid plane is balanced on a line connecting two supporting points a and b and is fixed by two springs c and d that have a summary stiffness coefficient of 1. Here and below, the green cylinders are corresponded to the weights of the criteria and the blue boxes are corresponded to the interaction indices of these criteria. We need these different shapes for visual separation between the weights corresponded to the criteria and the weights that reflect the interaction between the criteria.

The plane can rotate around the a–b support line and has no other freedom of movement. Plane rotation is limited and its deviation angle (marked "x" in Figure 2) from the horizon can change in [0, 1] boundaries. This angle is depicted on the aggregation scale that comprises an arc limited by values "0" and "1" in Figure 2.

In Figure 3, the symbols $g[1],\ldots,g[H]$ and $l[1],\ldots, l[K]$ on the plane indicate the lines along which the corresponding weights can move. On the left on the plane is the scale of the distances from the loads to the support line. This scale has limits from $-1$ to $1$.

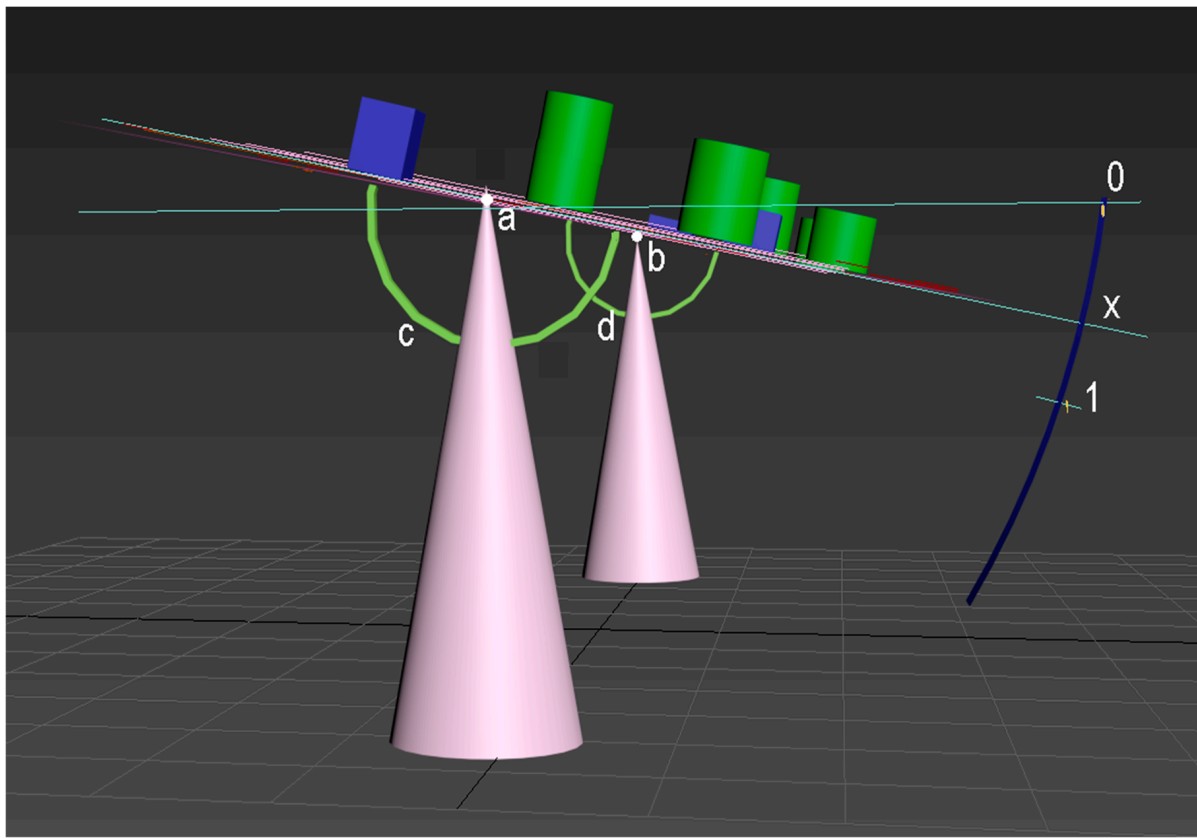

**Figure 2.** A side view of the 3D balance model.

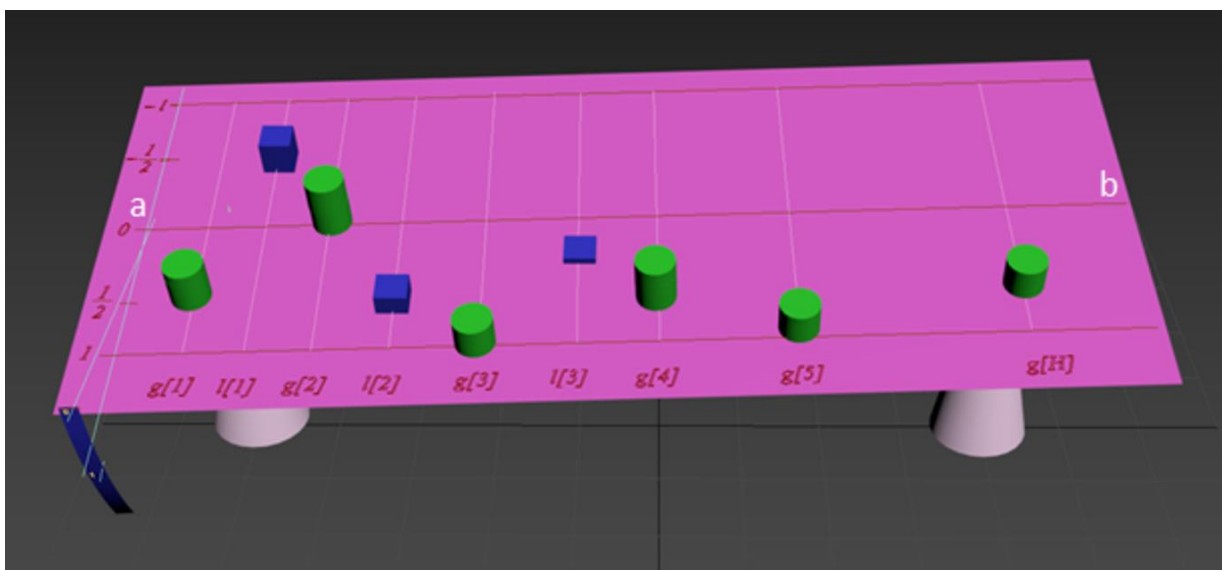

**Figure 3.** Another view of the 3D balance model.

Green cylindrical weights are placed on the part of the plane that is positioned on the aggregation-scale side of line a–b (Figure 3—bottom left) and their heights are directly proportional to weight values $\psi_1, \ldots, \psi_H$ or relative importance of criteria indexed $1, \ldots, H$. The distances from the support line to these cylinders are equal to the corresponding criteria values $g[1], \ldots, g[H]$. Auxiliary weights depicted by blue parallelepipeds that are placed on the plane at $l[1], \ldots, l[K]$ distances serve the purpose of depicting the influence of criteria dependent on the aggregation result, similar to the 2D visualization. The heights of these parallelepipeds are directly proportional to their weight values $\omega_1, \ldots, \omega_H$. The

distance $l[k]$ from weight $k$ to the support line is introduced by an established function $f_k^l$, the input values of which are criteria values and their weight values:

$$l[k] = f_k^l(g[1], \ldots, g[H], \psi_1, \ldots, \psi_H) \qquad (2)$$

We assume here that the decision maker can virtually move only green weights that correspond to different criteria and the auxiliary weights will move automatically according to Equation (2) for every $k$. The weight value $\omega_k$ of the $k$-th auxiliary weight is formed based on the decision maker's preferences with established function $f_k^\omega$, the input values of which are criteria weight values:

$$\omega_k = f_k^\omega(\psi_1, \ldots, \psi_H) \qquad (3)$$

Keeping in mind Newton's second law and understanding that the plane deviation angle is rather low ($cos x \approx 1$), the deviation angle value from the horizontal plane (aggregation result) is calculated in accordance with the equation below:

$$x = \sum_{h=1}^{H} \psi_h g[h] + \sum_{k=1}^{K} \omega_k l[k] \qquad (4)$$

In particular, if the visualized operator is a 2-order Choquet integral, each pair of interacting criteria $g[i]$ and $g[j]$, $\{i, j\} \subseteq J$ is associated with any single auxiliary weight:

$$l[k] = \begin{cases} \min(g[i], g[j]), & if \ I(i, j) > 0 \\ -\min(g[i], g[j]), & if \ I(i, j) < 0 \end{cases} \qquad (5)$$

where $I(i, j)$ is the $[i]$ and $g[j]$ interaction index, Equation (4) becomes (1) and this is equivalent to a 2-order Choquet integral. In case $I(i, j) = 0$, the corresponding auxiliary weight will not be present in the balance model. The proposed visualization helps to analyze predefined operator properties as well as to build operators with the desired properties by establishing Functions (2) and (3). In this case, the aggregation result under Equation (4) has the propensity to exceed the angles of plane rotation [0, 1]. In order to mitigate this contradiction, we assume that the influence of weights upon the plane which could potentially turn the plane beyond the limitation angle cannot turn it more than endlessly close to these limits. The formal equation is as follows:

$$x = \begin{cases} \sum\limits_{h=1}^{H} \psi_h g[h] + \sum\limits_{k=1}^{K} \omega_k l[k], \ if \ 0 \leq \sum\limits_{h=1}^{H} \psi_h g[h] + \sum\limits_{k=1}^{K} \omega_k l[k] \leq 1 \\ 0, \ if \ \sum\limits_{h=1}^{H} \psi_h g[h] + \sum\limits_{k=1}^{K} \omega_k l[k] < 0 \\ 1, \ if \ \sum\limits_{h=1}^{H} \psi_h g[h] + \sum\limits_{k=1}^{K} \omega_k l[k] > 1 \end{cases} \qquad (6)$$

This assumption (6) helps to implement weights in the 3D balance model, the summary weight values of which exceed 1 and are not equal to 1, as it is carried out in weighted average aggregation. Such weights can correspond to specifically derived criteria—for instance, with the "veto" effect [21]. As well as that, the model can be broadened by including functional relationships that affect the "lever scale sensitivity" [9].

Figure 4 shows a top view of the 3D balance model. For direct depictions of criteria interactions, they can be outlined by extra grey lines z and x connecting the weights (Figure 4). The blue parallelepiped connected with these lines to green cylinders is acting upon the plane like a coordinated result of criteria $g[1]$ and $g[2]$ interaction.

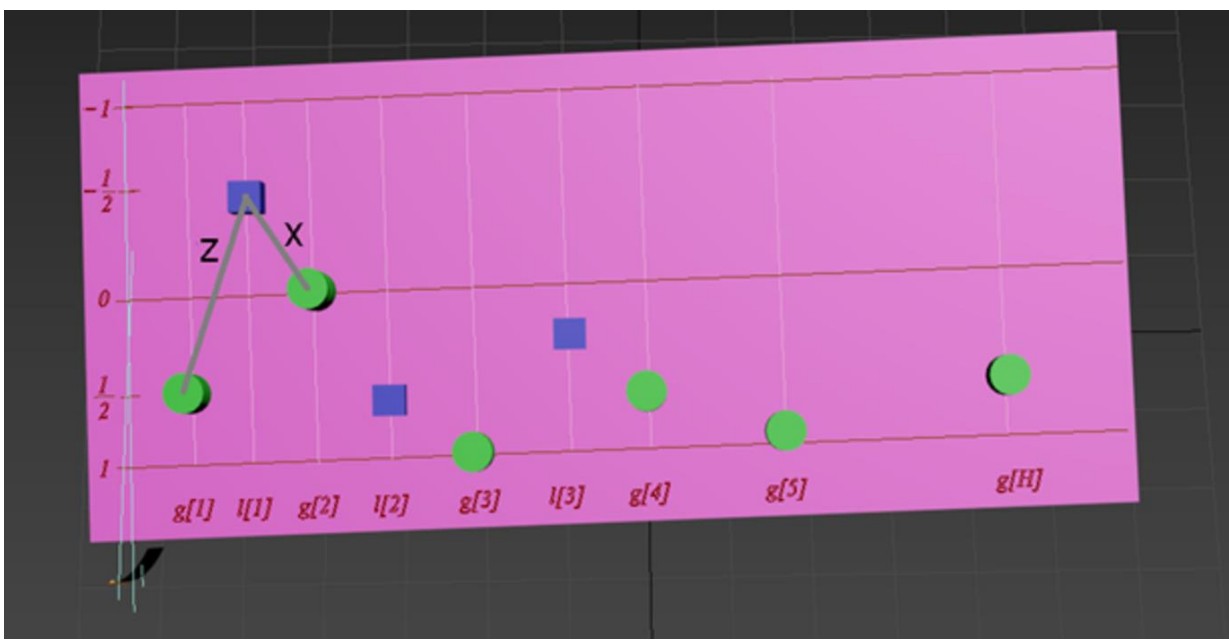

**Figure 4.** Top view of 3D balance model.

## 4. Aggregation-Operator Synthesis Using 3D Graphics

### 4.1. Criteria Choice and Normalization

A synthesized aggregation operator will reflect the point of view of the user (decision maker) concerning his preferences in a set of alternatives. This operator assigns to each alternative the result of aggregation on the unit interval [0, 1].

The initial aggregation phase requires a choice of aggregation criteria. Some of these criteria can be binary and reflect the presence or absence of any properties in a specific alternative. The other criteria part has quantitative character that is justified by the quantitative properties of the alternative. For instance, prices, distances, service levels and so on.

To be brief, we shall call normalized criteria attributes. Let us assume that for every alternative the decision maker knows the values of $y_1, \ldots, y_Y$ quantitative attributes and $z_1, \ldots, z_Z$ binary attributes, where $Y$ is the number of quantitative and $Z$ is the number of binary (Boolean) attributes. For example, in a hotel-choice task, the quantitative attributes $y_1, \ldots, y_Y$ can cover the different hotel properties: pricing, location (for instance, travel time to destination), comfort, room cleanliness, service quality. Binary attributes $z_1, \ldots, z_Z$ can be assigned as: beach availability, safety risks, hotel events that the tourist wants to visit under safe conditions.

In order to synthesize an aggregation operator, it is necessary to preliminarily normalize the quantitative and binary attributes by bringing them to a unit interval. As a result of such normalization, the aggregation criteria are achieved. A fully satisfactory alternative under a certain criterion for the decision maker will have a single value for this criterion which equals 1. And otherwise, if this alternative is totally unacceptable under a certain criterion, the criterion value will be zero. A full singular criterion value of 1 will correspond with a maximum input into the aggregation result while the zero value will represent a minimum influence.

There is no uniform approach to aggregation criteria normalization. More than 20 methods of such normalization exist, including linear and non-linear [26]. The normalization method choice can be handled under the assumption that it will bring a positive result in practical tasks [27].

It seems natural to utilize the whole unit interval for criteria variation in our 3D balance model. This is why our approach considers the linear method of max–min attribute normalization [27]. In accordance with this method, criteria normalization in our approach is realized through a procedure consisting of the following steps.

Step 1. Determine the domains of $y_1, \ldots, y_Y$ quantitative attributes as intervals limited by their minimal and maximum values $\min(y_1), \max(y_1), \ldots, \min(y_Y), \max(y_Y)$.

Step 2. Build equations $g[h] = \frac{1}{\max(y_h)}(y_h - \min(y_h))$, where $1 \leq h \leq Y$ for those aggregation criteria that have to increase together with the growth of their attributes.

Step 3. Build equations $g[h] = 1 - \frac{1}{\max(y_h)}(y_h - \min(y_h))$, where $1 \leq h \leq Y$ for those aggregation criteria that have to decrease in opposition to positive growth of their corresponding attributes.

Step 4. Build equations $g[h] = z_h$, where $Y + 1 \leq h \leq Y + Z$ for those binary attributes that, if equal to 1, lead to corresponding criteria values equal to 1. Determine which of these criteria equaling 1 lead to an aggregation result equaling 1 (the "favor" effect).

Step 5. Build equations $g[h] = 1 - z_h$, where $Y + 1 \leq h \leq Y + Z$ for those binary attributes that, if equal to 1, lead to corresponding criteria values equal to 0. Determine which of these criteria equaling 0 lead to an aggregation result equaling 0 (the "veto" effect).

The first step of this procedure determines the domains of the quantitative attributes such as pricing, location, comfort and so on. The second step forms aggregation criteria for which attribute growth leads to criteria increase. The third step forms aggregation criteria for which attribute growth leads to criteria decrease. The fourth and fifth steps define the binary criteria that will also play a role in the aggregation result. Some binary criteria may have a "veto" effect meaning that a zero value of a criterion leads to a zero aggregation result, while others—the opposite "favor" effect [28], meaning that if the criterion equals 1 then the aggregation result equals 1 with no consideration of other criteria values. As well as that, some criteria can be substituted, like price and comfort. In particular, such a conclusion can be reached by the decision maker on the basis of the fact that a hotel closer to the beach would be more expensive to stay at. Furthermore, the decision maker could demand that in case of safety hazards at the hotel, the aggregation result would zero out, and in the opposite case of no safety threats and an event of interest for the visiting tourist at the hotel—the aggregation result becomes 1. Such and similar criteria dependencies are convenient to analyze and verify using the 3D balance model.

### 4.2. Aggregation-Operator Synthesis Based on the 3D Balance Model

The suggested 3D balance model can be applied for different aggregation-operator synthesis including the Choquet integral. As mentioned earlier, the Choquet integral is well suited for multicriteria decision making because it allows one to model different interdependencies between criteria. This is the justification of our choice of it as the basis of aggregation-operator synthesis through visualization involving the 3D balance model.

As well as that, in order to use the Choquet integral, it is necessary to identify a fuzzy measure according to the decision maker's preferences. Such identification is troubled by the exponential difficulty growth of a fuzzy measure definition for each subset of criteria. Definition of all $2^H$ fuzzy measure coefficients is a rather hard or even an impossible task for the decision maker. Notice that even using three criteria, we have to define a fuzzy measure with $2^3 = 8$ coefficients. This is why M. Grabisch introduced the concept of k-additivity that helps to simplify the fuzzy measure by excluding criteria dependencies of more than $k$ criteria. In accordance with the abovementioned, we have chosen the 2-order Choquet integral that suits most practical cases [29].

In order to define a fuzzy measure, we have chosen the dispersion minimization method [30] among other methods [31] as, in contrast to other methods, it provides a uniqueness of the resulting fuzzy measure (or its absence, in the case of contradictions between the decision maker's preferences or their incompatibility with the Choquet integral properties) and a lack of subjectivism in the aggregation result apart from the subjectivism of the decision maker themself.

As a result, the 3D balance model will help us build the aggregation operator based on the Choquet integral with respect to 2-order fuzzy measure. The dispersion minimization method will be used for this fuzzy measure identification.

The initial balance model only has to reflect the aggregation criteria set without any additional information. Consequently, at this initial stage, we need to build a balance model that would include the chosen quantitative and binary criteria with equal weights, the criteria values should be neutral. For binary criteria, such a value would be zero, for quantitative 0.5. The aggregation result will correspond to the arithmetic mean and will also be equal to 0.5. Such a structure would be the initial point for further reasoning by the decision maker.

At the next phase, the binary criteria that possess the "veto" and "favor" effect need to be singled out, in order to examine them separately with the balance model and build the dependencies for the corresponding auxiliary weights.

After that, based on the input of the decision maker's preferences, the fuzzy measure with the minimal dispersion method needs to be identified, and the acquired fuzzy measure used to compile the Choquet integral and, finally, build the 3D balance model corresponding to this Choquet integral, considering the influence of the binary criteria. The decision maker could then check the resulting balance model for compliance with personal preferences by watching its output during criteria changes in virtual reality.

In the case of the model being compliant with the expert's preferences, the aggregation-operator synthesis is finished. Otherwise the decision maker has to return to the previous steps and change the structure.

Given the above considerations, we suggest the aggregation-operator synthesis procedure based on expert preferences using the 3D balance model. The input information for this procedure is the list of aggregated criteria formed by the normalization procedure described in the previous section and also the decision maker's preferences. The mentioned procedure consists of the following steps.

Step 1. Every quantitative criterion $g[h]$, $1 \leq h \leq Y$ receives a corresponding weight on the balance model. These weights are placed at 0.5 from the support line on the 3D balance model plane. Each binary criterion $g[h]$, $Y + 1 \leq h \leq Y + Z$ receives a corresponding weight on the balance model by placement on the support line of the plane.

Step 2. The binary criteria $g[h]$, $Y + 1 \leq h \leq Y + Z$, are examined for their involvement in the aggregation result. The "veto" and "favor" effect criteria are singled out. By considering the influence of the binary criteria on the aggregation result, the distance function $l[k]$ of the $k$ weight from the support line and its weight value $\omega_k$ is determined by adapting Equations (2) and (3) for binary criteria.

Step 3. The available realizations that were left over from the previous step are examined on a pair-by-pair basis and the question for which realization inside the pair the aggregation result would be higher should be answered; the partial weak order $\succsim_A$ on the set $A$ of available criteria realizations is then built. If possible, the desired values of the aggregation result should be defined for some criteria realizations from the list of several available realizations.

Step 4. The question, which criteria makes the most significant input into the aggregation result for each criteria pair, needs to be answered, and the corresponding preference relation or indifference relation entered to build a partial weak order $\succsim_J$ on the set $J = \{1, \ldots, H\}$ of criteria indices.

Step 5. Each criteria pair is examined for interaction (choosing the type and sign of interaction for each criteria pair, and pairs with similar, stronger or weaker interaction) and, on the basis of these conclusions, a partial weak order $\succsim_I$ is built on the set $I$ of pairs of criteria.

Step 6. Based on the partial weak orders $\succsim_A$, $\succsim_J$, $\succsim_I$ acquired at Steps 2–4, the fuzzy measure $\Psi$ is identified with the dispersion minimization method. The balance model for the acquired $\Psi$ fuzzy measure coefficients is built in accordance with Equations (4)–(6), taking into account the influence of binary criteria.

Step 7. The 3D balance model is examined for the resulting criteria weight values and auxiliary weight values corresponding with interaction indices at different criteria values. In the case of the 3D virtual realization not satisfying the decision maker, then a return

to Step 2 is required. If the need arises to add criteria to the criteria set, then, additional criteria should be formed by the normalization procedure and a return to Step 1 is required.

Now, we shall describe the suggested procedure step-by-step. At Step 1, aggregation criteria are visualized as corresponding weights in the 3D balance model. Simultaneously, the quantitative criteria receive "neutral" value 0.5 and binary criteria receive the value of zero that reflects no influence on the aggregation result. Realization of Steps 2–7 of this procedure has an iteration character that manifests itself in a step-by-step clarification of the decision maker's preferences that are in turn later used as input data for identifying fuzzy measures. This identification process uses the Kappalab software [28]. The Choquet integral with respect to the identified fuzzy measure represents the aggregation result without consideration of the influence of the binary criteria. The consideration of this influence is performed with weights corresponding to the binary criteria in the balance model.

## 5. Case Study

The tourist industry often deals with the task of choosing a hotel stay based on multiple criteria. The decision maker in this case may be a tourist, a travel agency manager providing tours, or both. The decision maker's preferences can be clarified and formalized with the implementation of the 3D balance model. Let us review an example of aggregation-operator synthesis reflecting tourist preferences for hotel choice.

During the first stage of this synthesis, it is necessary to choose hotel attributes that take part in this assessment. Research [32] represents the main 20 hotel attributes that provide this choice. Among these attributes we can define the most important ones: sleep quality, location, room quality, service, price, cleanliness [33]. These attributes were derived by means of studying text reviews by travelers from tripadviser.com which is one of the world's most popular web sites for travelers. On the one hand, the Choquet integral was implemented for modeling user preferences for hotel choices [15]. Keeping in mind that an average person can only keep track of seven objects in the operative memory simultaneously [34], we will limit this illustrative example of the 3D balance model to six criteria and will take the hotel attributes as described in [15,33]. We have chosen three generalized quantitative attributes: $y_1$—comfort, $y_2$—service, $y_3$—price. As well as that, we have chosen the following binary attributes, the values of which will be 0 for no and 1 for yes: $z_1$—attribute of presence or absence of services in the hotel provided with the use of intellectual technologies; $z_2$—attribute of safety hazard presence in the hotel; $z_3$—attribute of an event at the hotel that the tourist would like to visit if safety is ensured. The chosen attributes themselves can be presented as aggregation results of more detailed attributes. For example, attribute $y_1$ (comfort) can be presented as a result of aggregating such attributes as room quality, sleep quality, location, air conditioning, safe deposit box and so on. Thus a decision-making model can be built for the decision maker as a hierarchy of aggregation operators using several instances of the balance model, each of which serves the purpose of corresponding to aggregation-operator synthesis. This entire hierarchy of 3D balance models can be reflected in the same virtual reality. Our illustrational case corresponds with the top level of such a hierarchy.

In accordance with the above described procedure, the normalization of the chosen hotel attributes was realized.

At Step 1 of the procedure, the values $\min(y_1) = \min(y_2) = \min(y_3) = 0$; $\max(y_1) = \max(y_2) = \max(y_3) = 100$ were obtained. The result of the step is explained by the fact that the first two top level attributes have fuzzy character and therefore can be estimated in percent values, and the price (third attribute) can be estimated in universal units. Though it should be noted that lower hierarchy aggregation levels can include other scales for such parameters, such as the distance to the beach or the average waiting time at the bar.

At Step 2, the dependencies are built for those criteria where the growth of corresponding attributes leads to increases in aggregation results. These attributes are $y_1$ and $y_2$ because, according to the decision maker's reasoning, the higher the comfort and service

quality, the higher the hotel assessment result should be. The mentioned dependencies are as follows:

$$g[1] = \frac{y_1}{100} \tag{7}$$

$$g[2] = \frac{y_2}{100} \tag{8}$$

According to the decision maker, the higher the price per night, the lower the final hotel assessment shall be. Considering this, Step 3 of the normalization procedure constructs the equation:

$$g[3] = 1 - \frac{y_1}{100} \tag{9}$$

Steps 4 and 5 of the normalization procedure build dependencies for binary attributes $z_1$ and $z_2$. The decision maker's vision of this is the following. Attribute $z_1$ reflects the propensity of the decision maker to use "smart" technologies. Initially the decision maker considered attribute $z_1$, dealing with client interaction at the hotel using "smart" technologies to be binary, but clarified that they later started to envision this attribute as quantitative with a scale from 0% (total lack of communication with smart systems, only with human personnel) to 100% (interaction only with smart systems and no humans). Depending on the decision maker's opinion, this criterion can affect the aggregation result differently in terms of general hotel assessment. In particular, the use of "smart" technologies can facilitate lowering or total removal of tourist discrimination by race, sex, sexual orientation, social status [35]. In our case, the decision maker would like to minimize communication with human personnel, so for criterion $g[4]$, we can write:

$$g[4] = \frac{z_1}{100} \tag{10}$$

Attribute $z_2$ reflects the acceptability of personal safety risks in the hotel (0—no risks, 1—risks present). As these risks are unacceptable for the decision maker, so we can write for criterion $g[5]$:

$$g[5] = 1 - z_2 \tag{11}$$

At the same time, the zero value of criterion $g[5]$ corresponds with zeroing out the aggregation result ("veto" effect).

Attribute $z_3$ reflects the decision maker's motivation to visit an event held at the hotel under safety conditions (1—yes, 0—no):

$$g[6] = z_3 \tag{12}$$

Here, the value of 1 for criterion $g[6]$ leads to the corresponding aggregation result equaling 1 ("favor" effect).

In this case, normalization leads to a list of aggregation criteria that are calculated from the hotel attribute values according to Equations (7)–(12).

During the second synthesis stage, the aggregation operator for criteria $g[1], \ldots, g[6]$ is built using the aggregation-operator synthesis procedure from the previous section.

At Step 1 of the procedure, the quantitative criteria $g[1], g[2], g[3]$ are assigned with all equal weights in the 3D balance model at 0.5 distance from the support line (green cylinders in Figure 5, on the left). Auxiliary weights that serve the purpose of showing mutual dependencies of criteria for aggregation results are absent at this step. The binary criteria are assigned with gray-colored weights at the support line that are equal in mass to the green weights (Figure 5, on the right). We need it for visual separation between the quantitative criteria and the binary criteria.

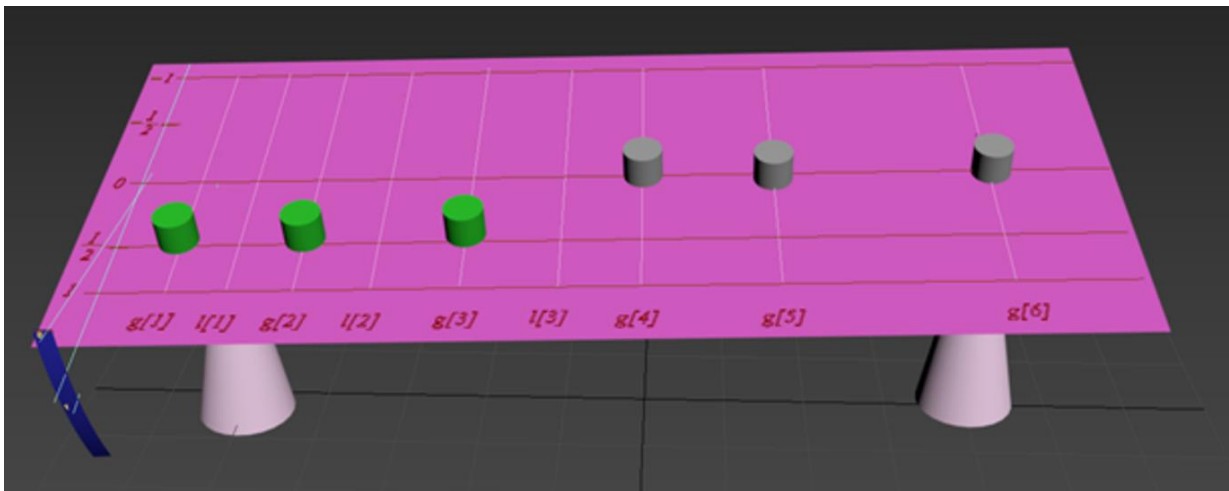

**Figure 5.** Initial view of the 3D balance model for criteria $g[1], g[2], g[3]$.

At Step 2 of the procedure, we examine the binary criteria that make decisions in the $\{0, 1\}$ segment. As the decision maker considers criterion $g[4]$ to be quantitative, this criterion shall be reviewed at Steps 2–7 of the procedure and the corresponding $g[4]$ weight shall be marked green as a quantitative criterion (Figure 6).

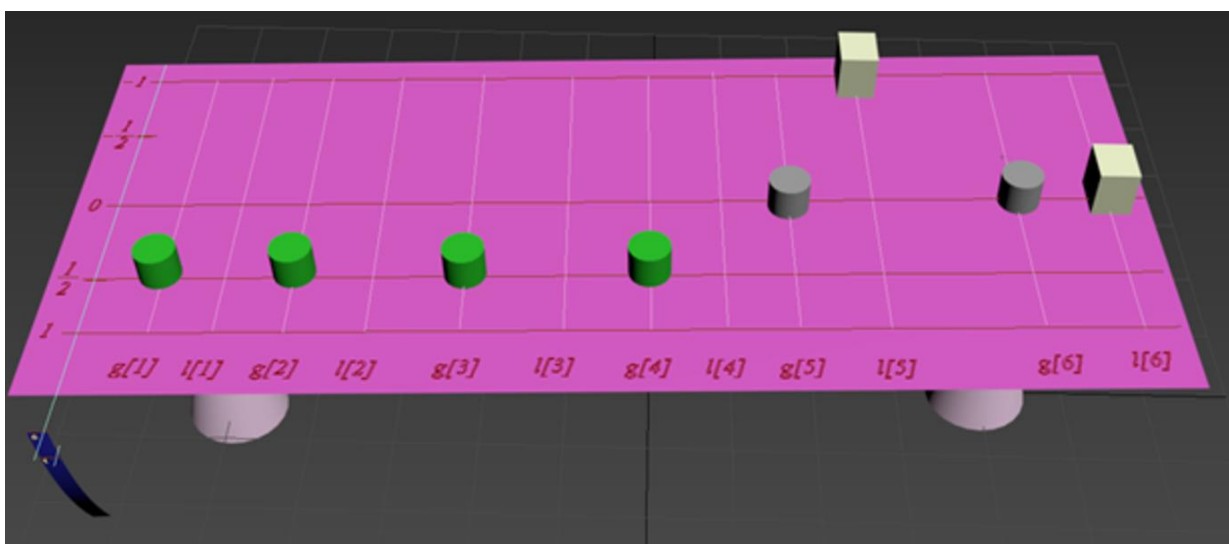

**Figure 6.** The 3D balance model after consideration of criteria $g[4], g[5], g[6]$.

It was already mentioned above that for the decision maker, any safety risks are unacceptable, and the corresponding $g[5]$ criterion should possess a "veto" effect and "overweigh", in the case of a zero value, all other influences on the plane. This is why the weight corresponding with criterion $g[5]$ is additionally provided with an auxiliary heavy enough weight $l[5]$ (gray parallelepiped on line $l[5]$ in Figure 6) allowing the result to be zeroed out in the case of $g[5]$ equaling zero: $l[5] = \begin{cases} 0, \; if \; g[5] = 1 \\ -1, \; if \; g[5] = 0 \end{cases}, \omega_5 = 2$. The weight value $\psi_5$ is assumed to be equal to zero as the influence of criterion $g[5]$ on the plane tilt is only driven by weight $l[5]$.

Criterion $g[6]$ of an intentional visit to an event at the hotel has a "favor" effect and "overweighs" all other influences on the plane in the case of a positive 1 value. This leads to providing the corresponding criterion $g[6]$ with a heavy auxiliary weight $l[6]$ (gray parallelepiped on line $l[6]$ in Figure 7) that forces a full 1 aggregation result in the case

$g[6]$ equals 1: $l[6] = \begin{cases} 0, \; if \; g[6] = 0 \\ 1, \; if \; g[6] = 1 \end{cases}$, $\omega_6 = 2$. The weight value $\psi_6$ is assumed to be equal to zero because the influence of criterion $g[6]$ on the plane tilt angle is conditioned only by weight $l[6]$. We need such a difference in colors and shapes of weights corresponding to separate different types of criteria for the convenience of their visual perception by an expert.

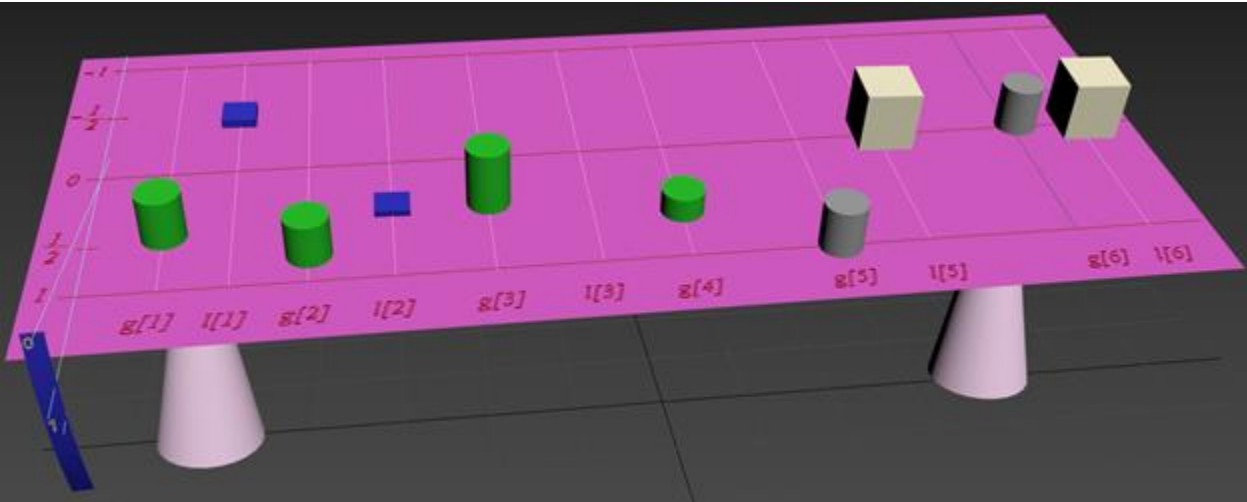

**Figure 7.** The 3D balance model after fuzzy measure identification.

At Step 3 of the procedure, the decision maker thinks in the following way. At the previous step, all criteria combinations were derived with $g[5] = 0$ or $g[6] = 1$, for which the aggregation result is preset without considering all other criteria values. So, the values of other criteria are worthy of reviewing only if $g[5] = 1$ and $g[6] = 0$. The available criteria realizations mean criteria values for those hotels that can be judged by the decision maker. This, for example, shows that the decision maker personally stayed at those hotels and personally gave marks for these criteria in hotel surveys. The available realizations in our case are provided in Table 1.

**Table 1.** Available criteria realizations.

| Hotel | Comfort | Service | Price | "Smart" Technologies | Safety | Event |
|:---:|:---:|:---:|:---:|:---:|:---:|:---:|
| | $g[1]$ | $g[2]$ | $g[3]$ | $g[4]$ | $g[5]$ | $g[6]$ |
| $a$ | 0.6 | 0.8 | 0.8 | 0.6 | 1 | 0 |
| $b$ | 0.3 | 0.4 | 1.0 | 0.5 | 1 | 0 |
| $c$ | 0.8 | 0.9 | 0.1 | 0.7 | 1 | 0 |

Hotel $a$ has average comfort and service levels in comparison with other hotels, satisfies the decision maker in terms of pricing a little more than average; the "smart" technologies criterion is a little higher than average and satisfies the decision maker. Hotel $b$ satisfies the decision maker most of all in terms of pricing compared to all known hotels but has relatively low comfort and service levels. Hotel $c$ provides a relatively high level of comfort and services and uses "smart" technologies but satisfies the decision maker only by 0.1 in terms of pricing.

The decision maker is hesitant to give each of the hotels a mark as the desired aggregation result and can only formulate personal preferences by a weak order for the set of available realizations. This weak order is as follows:

$$a \succ_A b \succ_A c \tag{13}$$

At Step 4 of the procedure, the issue of the relative importance of the criteria is considered for the quantitative criteria $g[1], \ldots, g[4]$. The decision maker acknowledges that the most unimportant criterion is the use of "smart" technologies $g[4]$. Among the remaining criteria, it is difficult for the decision maker to establish an importance range. The corresponding partial weak order over the set $J$ of criteria indices can be viewed as:

$$4 \prec_J 3; \ 4 \prec_J 1 \ ; \ 4 \prec_J 2 \tag{14}$$

At Step 5 of the procedure, the interactions between the criteria are considered. Here, the decision maker analyzes the quantitative criteria for their dependencies. And the decision maker thinks the following way at this step. In order to raise the general assessment of the hotel, it is sufficient to have either a high comfort level or a high service level. This is why the comfort criterion $g[1]$ and service criterion $g[2]$ have the effect of interchangeability and interact negatively.

The use of "smart" technologies in the tourist industry is currently only being tested for, as an example, deliveries of orders with autonomous vehicles [36] and is mostly devoted to tourist amusement. This is why the decision maker considers the use of "smart" technologies criterion not to be connected to other criteria and the interaction indices to be equal to zero.

Services are directly linked to pricing. Usually, the higher the price, the better the hotel service. This is why normalized (9) criteria $g[2]$ and $g[3]$ are negatively correlated and are interacting positively.

The conclusions provided by the decision maker can be extrapolated as signs of interaction indices for the corresponding criteria:

$$I(1,2) < 0; \ I(1,4) = I(2,4) = I(3,4) = 0; \ I(2,3) > 0 \tag{15}$$

At Step 6 of the procedure, fuzzy measure $\Psi$ is identified via the dispersion minimization method with limitations (13)–(15) obtained during earlier procedure steps. Based on this identification result, the weights corresponding to the criteria and the auxiliary weights related to the interaction indices are assigned with acquired weight values $\psi_h$ (Figure 7). Here, the weights on lines $g[1], g[2], g[3]$ correspond to the partial order (14), on line $l[2]$ at $g[3]$ distance from the support line is a weight corresponding to the interaction index of criteria $g[2]$ and $g[3]$. These criteria are negatively correlated and this corresponds to limitation (15). At this step on line $l[2]$ at distance $g[2]$ from the support line, there appears a weight that corresponds to interaction index $I(1,2)$ for criteria $g[1]$ and $g[2]$. This weight reflects substitutability or negative interaction between criteria $g[1]$ and $g[2]$ which is in line with the decision maker's conclusion about the low synergetic effect of these criteria. Specifically, for raising the general hotel assessment, it is in some way enough to have high comfort or high service level.

At Step 7, the decision maker watched the 3D balance model behavior during criteria value changes.

The decision maker did not consider it necessary to return to Steps 2 and 1 because the synthesized aggregation operator, the properties of which are reflected in the 3D balance model at this step, are well coordinated with the desired preferences.

This operator is viewed as follows:

$$P(g[1], \ldots, \ g[6]) = \begin{cases} 0, & if \ g[5] = 0 \\ 1, & if \ g[6] = 1 \\ C_\Psi(g[1], \ldots, \ g[4]) \ otherwise \end{cases} \tag{16}$$

Here, criteria $g[1], \ldots, g[6]$ are calculated according to Equations (7)–(12), $C_\Psi$ is the Choquet integral with respect to fuzzy measure $\Psi$ that was identified at Step 6 of the procedure. Fuzzy measure $\Psi$ is represented by the following coefficients (linked to the corresponding weight values of the model's weights): $\Psi_1 = 0.269$, $\psi_2 = 0.234$, $\psi_3 = 0.349$, $\psi_4 = 0.137$,

$\omega_1 = I(1,2) = -0.05$, $I(1,3) = 0$, $\omega_2 = I(2,3) = 0.05$, $I(1,4) = I(2,4) = I(3,4) = 0$. The aggregation results acquired with operator (16) for the available criteria realizations are in turn equal to $P(a) = 0.726$, $P(b) = 0.602$, $P(c) = 0.527$.

As expected, the synthesized aggregation operator (16) and its 3D balance model (Figure 7) after Step 6 reflect the decision maker's preferences expressed by limitations (13)–(15). This is proven, first of all, by the fact that the aggregation results for the available realizations are in order (13). Secondly, the limitation of the relative importance of the quantitative criteria (14) corresponds to the entirety of coefficients of fuzzy measures $\psi_1, \ldots, \psi_4$ and criteria interaction indices. Thirdly, the types of criteria interactions match the decision maker's preferences expressed by limitations (15).

Our illustrative case has shown the possibility of modeling the decision maker's preferences in virtual reality. If quantitative criteria values grow, the corresponding weights move away from the support line and increase their influence on the plane. At the same time, auxiliary weights devoted to quantitative criteria pairs that move behind the main weights can, depending on the interaction type, increase or decrease this influence.

Changes in the binary criteria values in the reviewed example lead to unconditional zeroing out ("veto" effect) or maximizing to 1 ("favor" effect) the plane tilt angle as the aggregation result, which is realized by heavy enough auxiliary weights "overweighing" the influence of all other criteria.

The reviewed case reflects the decision maker's opinion that mostly values money, then comfort and service, and considers, in particular, that in order to raise the overall hotel assessment mark, it is in some way necessary to have high comfort or a high service level—this is why these criteria interact negatively.

It was possible to formalize the other preferences as well. For example, a rich decision maker is not inclined to save money but values comfort and service and is not focused on using "smart" technologies and so on. For modeling such preferences, with the help of our approach, it is enough to set the corresponding limitations that do not match limitations (13)–(15) stipulated in the reviewed example.

## 6. Discussion

The implementation of this approach provided hotels ranging across several criteria with consideration of the decision maker's preferences reflected by an aggregation operator built upon the 2-order Choquet integral. Thanks to the use of the proposed 3D balance model, it became possible to visualize this aggregation operator and thus make it "visible" in virtual reality for the decision maker. By changing the criteria, the decision maker can watch the weights moving in the model and see the resulting changes in the tilt angle of the plane as the aggregation result. In the described balance model, the weights assigned to aggregation criteria have weight value and height that are directly proportional to fuzzy measure coefficients for these criteria. The aggregation-operator synthesis in our approach allows all available types of information concerning the decision maker's preferences to be used as input data: their personal preferences for the binary criteria influence on the aggregation result, the ranging of available alternatives by the decision maker, ranging of criteria importance, ranging of interaction indices for criteria pairs, interaction signs for criteria pairs as well as the desired aggregation operator values for available realizations. The implementation of the fuzzy measure identification method based on minimization of its dispersion enables the building of aggregation operators in the absence of certain types of the above-listed information without including additional subjectivity into the results apart from the personal subjectivity of the decision maker.

The design of this 3D balance model can be adjusted through the analysis of different user groups and usage variants [37]. For instance, by changing colors, weight forms, annotations and so on. In particular, in order to make decisions in the tourism industry, the criteria weights can be associated with symbols of comfort, service, price, safety, etc., [38]. The hierarchy of the aggregation operators and their corresponding balance models can be

shown in virtual reality as a nested multilevel structure that opens up, for example, if the cursor is put over the weight or symbol corresponding with the criterion.

In the scope of our approach, we reviewed the balance model with continuous and binary criteria as well as a continuous aggregation-result scale on the unit interval. This is explained by the fact that we wanted to demonstrate the top hierarchy level for decision making in the tourist industry with the example of a hotel assessment and choice task with continuous and binary criteria. Still, there are other options for creating balance models. Specifically, it is possible to assess criteria with discrete sets of marks on unit intervals as their function definition domains as well as to use a bipolar scale for the criteria themselves and for their aggregation result, which could be dictated by some other practical field of application. Also, the decision maker could demand that the sensitivity of the lever scale should change in a non-linear manner, as proposed in research [9]. Although the 3D balance model can facilitate the modeling of this non-linearity, we have, for now, reviewed only linear criteria scales and the linear max–min criteria normalization method [27]. Such a simplification is dictated by our desire for the decision maker to be capable of working with this approach after a short training course and having an intuitive vision of the aggregation operator and its synthesis process. The limitation of the proposed approach is the great visual complexity of constructing a three-dimensional balance model with more than seven criteria, which is caused by the nature of the human memory [34].

### 7. Conclusions

Revealing the key characteristics that significantly impact the tourist's decision making process and the synthesizing of the decision-making model by consumers could give hotel managers recommendations about which functions could be improved and how to attract tourists to stay at their hotel. An important part of such decision-making models are aggregation operators. Our goal was to empower the practicing specialists without mathematical education with a convenient and practical, easy-to-use approach to the analysis, synthesis and verification of aggregation operators. The proposed approach is based on a human-and-machine interaction in virtual reality through a 3D-graphics user interface. It provides opportunities to envision aggregation operator properties in the parameters of a virtual object that shares similar properties with a real physical object. By changing the virtual object parameters, the decision maker can synthesize an aggregation operator that would, in his opinion, best correlate with the correct estimation of alternatives by the aggregation of criteria values. A different decision maker would in turn watch the behavior of the 3D balance model during criteria changes and would analyze its compliance with their own personal preferences, including criteria interaction, and could correct this behavior and naturally the synthesized aggregation operator itself. Thus, based on a compromise formulation, it becomes possible to collectively synthesize and verify aggregation operators for multicriteria decision making in virtual reality [39].

Decision making using the proposed approach can be implemented by choosing one or another alternative, based on a comparison of the corresponding plane-deviation angles. As a result of the synthesis of the aggregation operator using a three-dimensional balance model, auxiliary weights may appear which correspond to hidden dependencies between the criteria that were not explicitly set by the expert. Thus, our approach will provide assistance in decision making, as well as allowing the generation of new knowledge based on the identification of hidden dependencies between the criteria.

Apart from the 2-order Choquet, integral 3D balance model visualization allows other aggregation operators to be built. Further studies could be connected with identifying, from among these operators, those that would be useful in a certain practical field, as well as methods for working with decision makers in order to define personal preferences and the synthesis of aggregation operators using the proposed visualization. In addition, using the proposed approach, it is possible to create new types of aggregation operators based on the formalization of the expert's preferences.

Another interesting research topic may be the synthesis of the proposed balance model based on data; in particular, by using the desired aggregation operator values as input data for available realizations. Such information can be obtained from surveys filled out by tourists. After this synthesis, the model could be presented to decision makers in order to estimate the correspondence of the acquired aggregation operator with personal preferences.

**Author Contributions:** Conceptualization, S.S.; methodology, S.S.; validation, A.A.; formal analysis, S.S.; investigation, S.S. and A.A.; resources, A.A.; writing—original draft preparation, S.S.; writing—review and editing, S.S. and A.A.; visualization, S.S.; supervision, S.S.; project administration, A.A.; funding acquisition, A.A. All authors have read and agreed to the published version of the manuscript.

**Funding:** The research (S. Sakulin and A. Alfimtsev) is carried out with the financial support of the Russian Science Foundation # 22-21-00711.

**Data Availability Statement:** No new data were created.

**Conflicts of Interest:** The authors declare no conflict of interest.

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
