# Peer review of "Multicriteria Decision Making in Tourism Industry Based on Visualization of Aggregation Operators"

_asi, doi:10.3390/asi6050074_

Round 1
Reviewer 1 Report
Dear Authors
Thank you for your advanced study. There are some of points which will modify the paper.
Regared,
Prof. Hamiden Khalifa

Author Response
We thank the Reviewer for their positive comments and useful suggestions.
- We have added a description of the advantages of this study. Advantages of the proposed approach are as follows. Firstly, it brings practical improvements in the formalization of expert preferences through the implementation of a virtual object that reflects the aggregation operator. Secondly, this approach can be applied with standard virtual reality software and hardware for the synthesis and verification of aggregation operators for decision making.
- We have improved the introduction section by adding links to a few recent articles in this area of aggregation operators. Recently, the authors of [36] proposed new aggregation operators based on Pythagorean cubic fuzzy sets for decision making. The paper [37] describes the use of extended families of OWA operators for efficient sentiment analysis in text reviews of tourists.
- We added the purpose of the study to the Abstract. The purpose of this study is to develop a decision-making approach that will allow practitioners to have a clear intuitive vision of the aggregation process and fuzzy measures.
- Incorrect symbols in the equations in Section 4 were found and corrected.
- The proposed approach allows us to model dependencies between more than two criteria in virtual reality. Thanks to the use of a three-dimensional balance model, it became possible to visualize aggregation operators and, thereby, make them “visible” in virtual reality for decision makers. Comparison with existing methods for visualizing aggregation operators is given in section 2.
The existing aggregation visualization methods encounter certain difficulties. Namely the fact that the graphical interpretation [8] is limited only by two criteria and the hierarchical diagram on its basis does not allow to “envision” the entire aggregation operator. Two-dimensional balance model [20] is not limited by two criteria but in case of multiple criteria (starting from about four to five – considering the added weights coordinated by interaction indices) the decision maker starts having trouble with a 2D visualization because the pictures of weights corresponding with interaction criteria and indices can overlap and cover each other. Apart from that, none of the mentioned visualization methods allows to model fuzzy measures higher than of second order that reflect dependencies of more than two criteria.
- The limitation of the proposed approach is the great visual complexity of constructing a three-dimensional balance model with more than 7 criteria, which is caused by the nature of human memory [32]. We have added a description of this limitation to Section 6 of the article.
- We have corrected the text of the article.
Reviewer 2 Report
The work deals with a subject of great interest and is of quality both in the methodology and in the bibliography.
However, almost the entire article is dedicated to explaining the methodology, but its application to the tourism sector, specifically to the hotel sector, is scarce, and it should be expanded a little more, indicating how the study was carried out, with which hotels and better explain the results.
The conclusions should also be expanded and link them more with the hotel sector.
Author Response
We thank the Reviewer for their positive comments and useful suggestions.
- Our article is devoted to the development of a theoretical approach in the field of decision-making, which can be applied in the field of tourism. Therefore, we did not set ourselves the goal of exploring specific hotels. Instead, we chose the most typical attributes of hotels from tourism publications and implemented an illustrative example of our approach using them. Our goal was to empower practicing specialists without mathematical education with a convenient and practical easy-to-use approach to analysis, synthesis and verification of aggregation operators. Further research will be devoted to the application of this approach to real data from the field of tourism.
Reviewer 3 Report
This paper has innovation and application value. To address the problem that the implementation of fuzzy integrals and fuzzy measures is limited by a weak intuitive understanding by practicing specialists of the aggregation process as well as fuzzy measures, the method of aggregation operator synthesis based on the 3D balance model and the 2-order Choquet integral was developed and applied in assessing and choosing a hotel.
However, there are some flaws in this paper.
1) Logical problem
For example, the connection between the second and third paragraphs of the introduction section is not smooth. The fourth and fifth paragraphs of the introduction section can be considered to be merged into one paragraph.
2)Incorrect sentences
Such as “Among these cases of research an important place belongs to aggregation operators based on fuzzy measures and fuzzy discrete integral.”
It’s suggested that this paper should be considered for publication after the textual expression has been carefully revised
Author Response
We thank the Reviewer for their positive comments and useful suggestions.
- We have added a sentence to the end of the second paragraph of the introductory section to make the connection to the next paragraph smoother. For such segmentation of tourists' preferences, "fine" decision-making methods can be applied. We have combined the fourth and fifth paragraphs of the introduction section.
- We have corrected the text of the article.
Reviewer 4 Report
1. The development of fuzzy is not clear, why they are fuzzy numbers and if other multicriteria methods can be used to provide better solutions to the tourism industry.
2. It is advisable to clarify more how the normalization results relate to the case study.
3. In the introduction, it is necessary to add the mathematical models in order to understand the importance they have.
4. Improve the conclusions to visualize as an aid to decision making and generate new knowledge.
Improve writing
Author Response
We thank the Reviewer for their positive comments and useful suggestions.
- Many studies of aggregation operators are devoted to fuzzy measures and integrals [23]. We used the Choquet integral with respect to fuzzy measures in our decision-making approach. Of course, other multi-criteria methods can be used to make decisions in the tourism industry. Nevertheless, fuzzy measures and integrals are widely used due to their flexibility and universality [23].
- We chose the Max-Min attribute normalization method [25] for our approach. In accordance with this method, an illustrative example describes the implementation of the normalization procedure. In particular, the criterion for the use of "smart" technologies in a hotel can help reduce or eliminate discrimination based on such tourist characteristics as race, gender, sexual orientation, and social status [33].
- We have improved the introduction section by adding links to several recent articles describing various mathematical models.
- We have added conclusions in Section 7 about the use of visualization as an aid to decision making and generating new knowledge.
Reviewer 5 Report
The paper offers an intriguing and innovative discourse within the realm of tourism studies, and certainly stands as a noteworthy contribution to the field. The authors have conducted a thorough and comprehensive review of the existing literature, meticulously sifting through previous works to contextualize their own research.
The paper's methodological design is well detailed and systematically laid out, even though I must admit that I am not profoundly versed in the specific type of methodology applied in this study. Nonetheless, it's evident that a great deal of thought and planning has been put into the research design, which contributes to the overall rigor of the study.
The way the authors manage, interpret, and present the data is very lucid and articulate, making it easy for readers to follow and comprehend their analysis. They do a commendable job in taking complex data and translating it into accessible, digestible information.
While the paper is remarkably well-structured, I would recommend that the authors take another look at the conclusions section. It would be beneficial to have more elaboration on the broader implications of their research findings. In other words, how does their work contribute to the existing body of knowledge in the field, and what practical applications might it have? Reflecting on these questions and addressing them in the conclusion could significantly enhance the paper's overall impact.
Author Response
We thank the Reviewer for their positive comments and useful suggestions.
We have added conclusions to Section 7 about the use of visualization as an aid to decision-making and the generation of new knowledge, as well as to create new aggregation operators.
Round 2
Reviewer 3 Report
The paper has been improved accordingly.
Author Response
We thank the Reviewer for their positive feedback.

Reviewer 4 Report
The manuscript is well structured despite the fact that it has few bibliographical references.
It is necessary to reduce the number of keywords, there are too many. Select the most important.
Minor editing of English language required
Author Response
We thank the Reviewer for their positive comments and useful suggestions.
We have added some bibliographical links and removed unnecessary keywords.
Final editing of the English language will be done before publication.
